# Risk factors contributing to infection with SARS-CoV-2 are modulated by sex

**Octavio A. Lecona**[1,2], **América G. Arroyo-Valerio**[3], **Nallely Bueno-Hernández**[3], **José Damian Carrillo-Ruíz**[3,4], **Luis Ruelas**[1,2], **René Márquez-Franco**[1,2], **Alejandro Aguado-García**[3,5], **Eira Valeria Barrón**[6], **Galileo Escobedo**[3], **Elizabeth Ibarra-Coronado**[2,7], **Paola V. Olguín-Rodríguez**[2,5], **Antonio Barajas-Martínez**[2,7], **Ana Leonor Rivera**[2,8], **Ruben Fossion**[2,8]*

**1** Doctorado en Ciencias Biomédicas, Universidad Nacional Autónoma de México (UNAM), Mexico City, Mexico, **2** Centro de Ciencias de la Complejidad (C3), Universidad Nacional Autónoma de México (UNAM), Mexico City, Mexico, **3** Dirección de Investigación, Hospital General de México Dr. Eduardo Liceaga, Mexico City, Mexico, **4** Coordinación de Neurociencias, Facultad de Psicología, Universidad Anahuac México, Mexico City, Mexico, **5** Centro de Investigación en Ciencias (CInC), Universidad Autónoma del Estado de Morelos, Cuernavaca, Morelos, Mexico, **6** Servicio de Medicina Genómica "Hospital General de México Dr. Eduardo Liceaga, Mexico City, Mexico, **7** Facultad de Medicina, Universidad Nacional Autónoma de México, Mexico City, Mexico, **8** Instituto de Ciencias Nucleares, Universidad Nacional Autónoma de México, Mexico City, Mexico

* ruben.fossion@nucleares.com

**Data Availability Statement:** All relevant data are within the paper and its Supporting information files.

## Abstract

Throughout the early stages of the COVID-19 pandemic in Mexico (August—December 2020), we closely followed a cohort of n = 100 healthcare workers. These workers were initially seronegative for Immunoglobulin G (IgG) antibodies against SARS-CoV-2, the virus that causes COVID-19, and maintained close contact with patients afflicted by the disease. We explored the database of demographic, physiological and laboratory parameters of the cohort recorded at baseline to identify potential risk factors for infection with SARS-CoV-2 at a follow-up evaluation six months later. Given that susceptibility to infection may be a systemic rather than a local property, we hypothesized that a multivariate statistical analysis, such as MANOVA, may be an appropriate statistical approach. Our results indicate that susceptibility to infection with SARS-CoV-2 is modulated by sex. For men, different physiological states appear to exist that predispose to or protect against infection, whereas for women, we did not find evidence for divergent physiological states. Intriguingly, male participants who remained uninfected throughout the six-month observation period, had values for mean arterial pressure and waist-to-hip ratio that exceeded the normative reference range. We hypothesize that certain risk factors that worsen the outcome of COVID-19 disease, such as being overweight or having high blood pressure, may instead offer some protection against infection with SARS-CoV-2.

## Introduction

In a recent publication, we reported on results of a cohort of healthcare workers (HCW) from the Hospital General de México Dr. Eduardo Liceaga (HGMEL), who were exposed during the

**Funding:** This research was funded by the Consejo Nacional de Humanidades, Ciencia y Tecnología (CONAHCYT México) with grant number 312512 to JDCR, and the CONAHCYT-FORDECYT-PRONACES program with grant number 610285/2020, and grant number 263377/2020 to ALR. Funding was received also from the Dirección General de Asuntos del Personal Académico (DGAPA) from the Universidad Nacional Autónoma de México (UNAM) with grant number PAPIIT IN110321 to RF. OAL is a doctoral student from the Programa de Doctorado en Ciencias Biomédicas from the Universidad Nacional Autónoma de México (UNAM) and has received fellowship 697939 from CONAHCYT México. We are also grateful for financial support from the Research Division of HGMEL with grant number DI/20/501/04/32 to AGAV, NBH, JDCR, EVB, and GE. The funders had no role in study design, data collection and analysis, decision to publish, or preparation of the manuscript.

**Competing interests:** The authors have declared that no competing interests exist.

early phase of the COVID-19 pandemic in Mexico, from August to December 2020 [1]. The cohort underwent a baseline evaluation during which a variety of categorical and continuous variables were registered, including medical history, demographic, anthropomorphic, clinical and laboratory information, and working conditions. Possible infection with SARS-CoV-2 was detected at baseline, using an anti-SARS-CoV-2 Immunoglobulin G (IgG) antibody serum titers test and a quantitative polymerase chain reaction (qPCR) test. Possible infection was examined as well during the evaluation period in symptomatic participants using a qPCR test, and during a follow-up study six months later using an IgG test. The cohort included only participants who tested negative at baseline and was analyzed by groups defined post-hoc depending on the outcome of the IgG test at 6 months, using a group that remained seronegative (IgG-) and a group that became seropositive (IgG+). This previous publication had multiple study objectives. The first objective was to evaluate the incidence of SARS-CoV-2 infection in HCW of HGMEL, and to compare results from the qPCR and IgG tests. The incidence rate of SARS-CoV-2 infection was considerably higher in our cohort (58%) than in other healthcare populations as published by other authors (5–29%) [2–9]. Most reports used qPCR tests to detect positive cases. Since qPCR amplifies the genetic material of the virus, this methodology can only identify active infection cases, which may lead to underestimating asymptomatic patients and incidence rates among healthcare professionals. Therefore, one of our conclusions was that the incidence of SARS-CoV-2 infection in HCW should be assessed by IgG antibody measurements irrespective of having presented COVID-19 symptoms or not. The second objective was to analyze the incidence rate of infection in HCW depending on the workplace in the hospital. We found that HCW who were correctly wearing personal protection equipment (PPE), exhibited infection rates that were similar for high-exposure (e.g., emergency room or intensive care units) and moderate- and low-exposure areas (e.g., hospital administration or medical social work). In correspondence with previous studies [10, 11], we concluded that the risk of SARS-CoV-2 infection in HCW is not necessarily associated with the workplace but rather with other factors such as the inaccurate use of PPE, and community exposure or contact with family members who are asymptomatic SARS-CoV-2 carriers. A third objective was to evaluate the prevalence of symptoms related to SARS-CoV-2 infection. We found that only 33% of IgG seropositive participants exhibited symptoms, whereas in other studies asymptomatic carriers among HCW varied from 4.8 to 40% [12, 13]. These results indicate that the relatively high percentage of asymptomatic SARS-CoV-2 carriers among HCW may have contributed to the large transmissibility of the disease both in the hospital and in the community. Finally, the last objective of our previous publication—which is also the study subject of the present contribution—was to evaluate demographic and anthropometric characteristics and laboratory parameters measured at the baseline in the cohort as risk factors for infection with SARS-CoV-2 over a time period of six months. In line with previous studies [10, 11, 14], no specific risk factors could be identified between any of the categorical or continuous variables of the dataset. The only exception was the presence of hypertension at baseline in participants that would remain IgG seronegative, and absence of hypertension in participants that would become seropositive. No additional information was recorded for the hypertensive participants (e.g., the degree of hypertension), but all hypertensive participants were medicated and had blood pressure measurements in the range normal to normal-high, indicating that their hypertensive condition was under control. We concluded that it is possible that participants recognizing themselves as a vulnerable population with comorbidities, such as hypertension, followed safety procedures and social distancing more strictly than participants without known comorbidities, and therefore by adapting their behavior were less likely to become infected [1].

The purpose of the present contribution is to reanalyze the dataset of demographic and anthropometric characteristics and laboratory parameters of the cohort using more advanced statistical methods. Our previous contribution [1] used common univariate statistical analyses, such as the Student t-test for continuous variables and the $\chi^2$-test for categorical variables, and the conclusions on risk factors for infection with SARS-CoV-2 are counterintuitive because hypertension was defined early on during the pandemic as an important risk factor for adverse health results in the development of the COVID-19 disease. The results of our previous publication were based on a preliminary dataset of n = 110 participants that was constructed with the parameters and the evaluation results that were available at that time. Continuous variables included breathing rate (BR), temperature, weight, body mass index (BMI), waist circumference, hip circumference, blood glucose, urea, creatinine, uric acid and cholesterol. Categorical variables included obesity, type-2 diabetes and hypertension. The present contribution uses an updated dataset that includes additional parameters such as heart rate (HR), mean arterial pressure (MAP) and waist-to-hip ratio (WHR), but only considers n = 100 participants with complete data. It has been argued before that susceptibility to infection might stem from systemic and physiological factors, rather than solely from local or cellular properties [15]. We hypothesized that a multivariate statistical approach, that considers interactions between different categorical variables or factors, and evaluates the combined effect of multiple continuous variables, better reflects the fact that the human body is a system where physiological variables do not exist in isolation, and therefore might detect differences in susceptibility to infection where univariate analyses do not. For this reason, the aim of the present study is to analyze the dataset of the cohort at baseline using multivariant statistical approaches, such as MANOVA, to identify risk factors for infection with SARS-CoV-2.

## Methods

### Trial design and ethical aspects

This prospective cohort study involved the enrollment of various professionals, including doctors, nurses, researchers, clinical laboratory technicians, psychologists, rehabilitators, and administrative staff at HGMEL, one of the largest hospitals of the Ministry of Health in Mexico City, designated for COVID-19 patient care during the pandemic. Participant enrollment occurred between August 18 and October 2, 2020, while the follow-up period of six months extended from October 2, 2020, to March 31, 2021. All participants provided a written informed consent. This study adhered to the guidelines set forth in the Strengthening the Reporting of Observational Studies in Epidemiology (STROBE) statement for reporting observational studies. The study received approval from the Ethics and Clinical Research committees of HGMEL (Approval No: DI/20/501/04/32) and was conducted in strict accordance with the principles outlined in the 1964 Declaration of Helsinki and its subsequent amendment in 2013.

### Sample size

In our previous publication [1], we described the method for calculating the sample size, which was based on the number of HCW (medical doctors, nurses, researchers, psychologists, and rehabilitators) who continued working throughout the pandemic. Of the initial 7000 healthcare personnel working at HGMEL, 2000 of the healthcare workers with highest risk factors decided to withdraw during the pandemic, leaving a total of $N$ = 5000 personnel who continued working, representing the total size of the active population. The determination of

sample size by estimation of a proportion was estimated using the following expression [16]:

$$n = \frac{z^2 pq}{e^2 + \frac{z^2 pq}{N}}$$

where $n$ is the required sample size, $z$ is the critical value of the standard normal distribution associated with the desired confidence level, $p$ is the expected estimate of the proportion of infected individuals in the population, $q$ is the complement of $p$ ($q = 1—p$), $e$ is the allowable margin of error in the proportion estimation, $N$ is the total size of the population. Based on previous studies [2, 3], where 13% of the workers are infected with the SARS-CoV-2 virus with a margin of error of 10% and a 99% confidence level, the estimated value for the required sample size is $n = 74$.

## IgG seroprevalence analysis

All participants underwent blood sampling at the beginning of the study (August 2020) and again six months later (January 2021) to determine the seroprevalence of the presence of antibodies against SARS-CoV-2. Specific IgG antibody levels against the SARS-CoV-2 nucleocapsid (N) protein were measured in triplicate by the Enzyme Linked-Immuno Sorbent Assay (ELISA) by a standardized kit from Abcam (Abcam, ab274339, Cambridge, UK), using a microplate reader at 450 nm.

Most participants in this study (85 out of 100, equating to 85%) received their SARS-CoV-2 vaccinations beginning in December 2020, notably after the commencement of the study, and continuing until the follow-up evaluation in January 2021. In all cases, the mRNA-based Pfizer–BioNTech COVID-19 vaccine (brand name Comirnaty) was used. ELISA differentiates between immune responses to infection with SARS-CoV-2 and immune responses caused by all vaccines against COVID-19 that were approved at that time [17]. There is a consensus that vaccination against COVID-19 offers protection against severe illness, whereas the protection against the infection itself is minimal and short-lived [18, 19].

## Statistical analysis

MANOVA is a statistical technique used to simultaneously analyze two or more continuous dependent variables between two or more groups, and considers the interactions among independent categorical variables, provides insights into multivariate relationships that may not be evident when examining each variable separately, and enhances the understanding of the underlying relationships by considering the joint variation among variables [20, 21]. However, MANOVA relies on certain assumptions, such as normality of the data and absence of multi-collinearity between variables, and violations of these assumptions can affect the validity of the results [22]. In this study, we employ MANOVA to examine relationships between variables and group differences, aiming to capture the complex interplay among variables and obtain a comprehensive understanding of their associations.

The dataset comprises multiple categorical and continuous variables. To construct a multivariate model, we considered that many of the categorical variables could be implicitly incorporated using continuous variables; for example, diabetes can be accounted for by blood glucose, hypertension by mean arterial pressure, and being overweight or having obesity by body mass index (BMI). Variables potentially exhibiting multi-collinearity, such as weight and height, or systolic and diastolic blood pressure, were excluded, and instead combined variables such as BMI and mean arterial pressure (MAP) were used. However, the categorical variable of sex cannot be represented by any continuous variable and was explicitly included in the model. Therefore, we built a MANOVA model with two factors: Susceptibility, i.e., whether

participants remain seronegative or become infected with SARS-CoV-2 at 6 months (IgG- vs. IgG+), Sex (male M vs. female F), and their interaction (Susceptibility × Sex). This results in four groups: male and female participants at baseline who remain seronegative (M- and F-), or who become infected (M+ and F+). Statistical significance was set at the $p<0.05$ level.

To conduct a comprehensive analysis of the differences between groups, several statistical procedures were performed. Firstly, a logarithmic transformation was applied to the non-normally distributed variables, including urea, creatinine, uric acid, blood glucose, BMI, WHR, HR, BR, body temperature, and MAP. These variables were transformed to meet the assumptions of parametric analyses. Cholesterol was the only variable that exhibited a normal distribution for all groups, and therefore did not require a logarithmic transformation.

Subsequently, a parametric multivariate analysis of variance (2-way MANOVA) was conducted to assess the main effects of the factors of Susceptibility and Sex, and the interaction Susceptibility × Sex on the combined continuous variables. Additionally, a parametric univariate analysis of variance (2-way ANOVA) was performed to study the main effects and the interaction for each of the continuous variables individually.

Furthermore, to complement the parametric analyses, the non-parametric, univariate and pairwise Mann-Whitney U test was applied. This test was used to evaluate the differences in individual variables between groups without making assumptions about the underlying distribution of the data.

All analyses were performed using Statistical Package for the Social Sciences (SPSS) software version 22.0.

## Results

Originally, $n = 115$ healthcare workers were included; however, for this study, data from only $n = 100$ participants were considered due to inclusion criteria and incomplete data. Fig 1

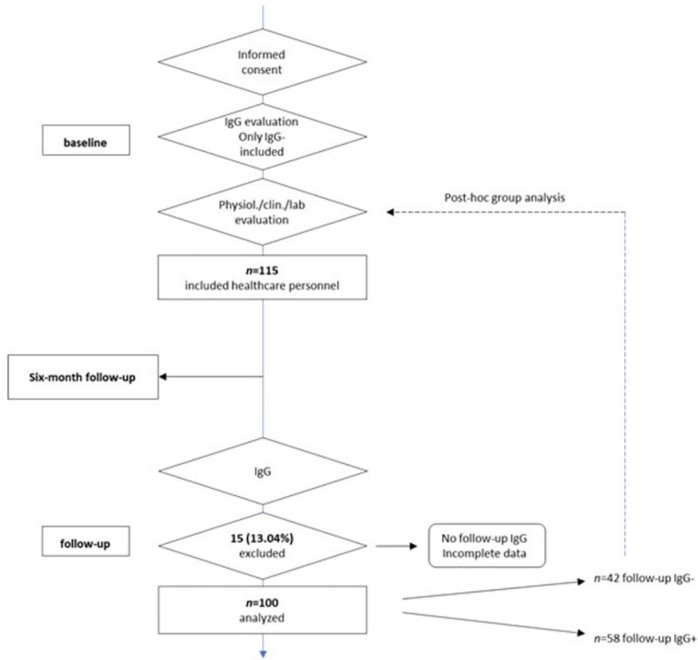

**Fig 1. Flowchart of the selection process of the participants based on the inclusion and exclusion.**

shows a flowchart of the selection process. The composition of the cohort was delimited by the factors Sex and Susceptibility for infection over the course of 6 months, see Table 1.

All variables, except cholesterol, exhibited non-normal distributions in at least one of the groups. Therefore, values for these variables were reported in Table 2 in the format Q2 (Q1, Q3), where Q2 represents the second quartile (i.e., the median), and Q1 and Q3 represent the lower and upper quartiles, respectively. Parametric analyses such as MANOVA and ANOVA require data to be normalized, which can be achieved by using a logarithmic transformation. All non-normal data were transformed using a base-10 logarithmic function ($\log_{10}$) and are reported in the format mean (SD), where SD is the standard deviation. Cholesterol, on the other hand, is reported in both formats without applying a log transform.

A 2-way MANOVA analysis was conducted to examine the main effect of factors Susceptibility, Sex and their interaction. MANOVA calculates a score that quantifies the combined effect on all continuous variables, see (Fig 2). There was no statistically significant main effect of Susceptibility on the combined dependent variables, $F_{(11, 86)} = 1.749$, $p = 0.076$, Wilks $\Lambda = 0.817$; there was a statistically significant main effect of Sex on the combined dependent variables, $F_{(11, 86)} = 4.603$, $p<0.001$, Wilks $\Lambda = 0.629$; there was also a statistically significant interaction effect between Susceptibility and Sex on the combined dependent variables, $F_{(11, 86)} = 1.967$, $p = .042$, Wilks $\Lambda = .799$.

After conducting a MANOVA analysis to examine systemic differences, 2-way ANOVA analyses were performed to examine effects of the main factors Susceptibility, Sex and their interaction for each of the individual continuous variables, see Table 2. The results indicated a significant main effect for Susceptibility on urea, $F_{(1, 96)} = 5.199$, $p = 0.025$, partial $\eta^2 = 0.051$; and also a significant main effect for Sex, $F_{(1, 96)} = 17.160$, $p<0.001$, partial $\eta^2 = 0.152$; but no significant interaction between Susceptibility and Sex, $F_{(1, 96)} = 0.803$, $p = 0.372$, partial $\eta^2 = 0.008$. There was no significant main effect for Susceptibility on creatinine, $F_{(1, 96)} = 0.291$, $p = 0.591$, partial $\eta^2 = 0.003$; but there was a significant main effect for Sex, $F_{(1, 96)} = 13.661$, $p<0.001$, partial $\eta^2 = 0.125$; and also a significant interaction between Susceptibility and Sex, $F_{(1, 96)} = 4.733$, $p = 0.032$, partial $\eta^2 = 0.047$. There was no significant main effect for Susceptibility on uric acid, $F_{(1, 96)} = 2.364$, $p = 0.127$, partial $\eta^2 = 0.224$; but there was a significant main effect for Sex, $F_{(1, 96)} = 10.944$, $p = 0.001$, partial $\eta^2 = 0.102$; and no significant interaction between Susceptibility and Sex, $F_{(1,96)} = 3.256$, $p = 0.074$, partial $\eta^2 = 0.033$. There was no significant main effect for Susceptibility on WHR, $F_{(1, 96)} = 3.624$, $p = 0.060$, partial $\eta^2 = 0.036$; but there was a significant main effect for Sex, $F_{(1, 96)} = 15.147$, $p<0.001$, partial $\eta^2 = 0.136$; and also a significant interaction between Susceptibility and Sex, $F_{(1, 96)} = 13.235$, $p<0.001$, partial $\eta^2 = 0.121$. There was no significant main effect for Susceptibility on MAP, F

**Table 1. Composition of the cohort in terms of male (M) and female participants (F), and post-hoc defined groups of participants that remain seronegative (IgG-) or become seropositive (IgG+) over the course of 6 months.**

| Factors and interaction | | n |
|---|---|---|
| Susceptibility | IgG- | 42 |
| | IgG+ | 58 |
| Sex | F | 65 |
| | M | 35 |
| Susceptibility × Sex | F- | 29 |
| | F+ | 36 |
| | M- | 13 |
| | M+ | 22 |

**Table 2. Baseline values for anthropometric, biochemical, and physiological variables of the groups of male and female participants that remain seronegative (M-and F-, respectively) and become seropositive over the course of 6 months (M+ and F+, respectively).** All variables, except cholesterol, are non-normal for at least one of the groups, and therefore are reported in standard units as Q2(Q1, Q3) (upper half of the table), where Q2 is second quartile, i.e., the median, Q1 y Q3 are first and third quartiles respectively. Non-normal data is log transformed and represented as mean (SD) (bottom half of the table), where SD is standard deviation. For completeness, cholesterol is given in both formats without log transformation. Statistically significant factors and/or interaction using 2-way ANOVA and statistically significant pairwise group comparisons using the Mann-Whitney U test are also indicated for individual variables.

| Variable | Reference range | | M- | M+ | F- | F+ | Significant factors for 2-way ANOVA | Clinical criteria reference |
|---|---|---|---|---|---|---|---|---|
| | Male | Female | (n = 13) | (n = 22) | (n = 29) | (n = 36) | | |
| **urea (mg/dL)** | 10–50 | 10–50 | 39.3(33.5,45)[#],[§] | 34.7(29.5, 38.3)[§] | 28.2 (25.55,34.55) | 28.3(22.45, 32.55) | – | [23] |
| **creatinine (mg/dL)** | 0.8–1.5 | 0.5/1.1 | 0.93(0.83,1.05)[#],[§] | 0.82(0.76, 0.92)[#] | 0.75(0.68, 0.86) | 0.75 (0.68,0.87) | – | [24] |
| **uric acid (mg/dL)** | <6.8 | <6.8 | 5.8(5.68,6.82)[¶],[#],[§] | 5.55(4.6, 6.2) | 4.6(4.28, 5.53) | 4.45(4.2,5.65) | – | [25] |
| **glucose (mg/dL)** | 70–100 | 70–100 | 94(84,103) | 89.5(85,96) | 87(82.5,102) | 89(82.5,93.5) | – | [26] |
| **BMI (kg/m$^2$)** | 18–27 | 18–27 | 27.7(26.17,29.35) | 26.1 (23.15,29.39) | 27.3(24.2,29) | 26.4 (23.6,28.7) | – | [27] |
| **WHR** | 0.78–0.94 | 0.71–0.84 | 0.95(0.91,1)[¶],[#],[§] | 0.86(0.83, 0.91) | 0.83(0.78, 0.88) | 0.87(0.82,0.9) | – | [28] |
| **temperature (˚C)** | 36.2–37.5 | 36.2–37.5 | 36.3(36.15,36.4) | 36.3(36,36.4) | 36.1(35.98,36.3) | 36.2 (36.1,36.4) | – | [29–31] |
| **MAP (mmHg)** | 70–93 | 70–93 | 107.3 (99.4,116.6)[¶],[#],[§] | 96.67(85.33, 104.33) | 92.67 (86.42,98.92) | 94.5(88.5, 104.5) | – | [32] |
| **HR (bpm)** | 65.4–83.2 | 70.9–87.3 | 73(68.75,77) | 74.5(66,83) | 75(68.75,79.25) | 74.5(69.5,81) | – | [33] |
| **BR (cpm)** | 12–28 | 12–28 | 15(13.75,16) | 13.5(12,17)[#] | 16(14,20) | 15(13,16) | – | [34] |
| **cholesterol (mg/dL)** | <200 | <200 | 179(162, 198) | 175 (157,205) | 193 (159.3,204.3) | 182(153,206) | – | [35] |
| **urea (log$_{10}$ mg/dL)** | – | – | 1.6(0.12) | 1.52(0.09.) | 1.47(0.126) | 1.44(0.115) | **Sex, Sus**. | – |
| **creatinine (log$_{10}$ mg/dL)** | – | – | -0.03(0.064)[#] | -0.08(0.061) | -0.13(0.072) | -0.1(0.088) | **Sex, Sus. × Sex** | – |
| **uric acid (log$_{10}$ mg/dL)** | – | – | 0.79(.084) | 0.72(0.095) | 0.68(0.08) | 0.69(0.106) | **Sex** | – |
| **glucose (log$_{10}$ mg/dL)** | – | – | 1.99(0.09) | 1.95(0.047) | 1.95(0.051) | 1.95(0.056) | - | – |
| **BMI (log$_{10}$ kg/m$^2$)** | – | – | 1.45(0.055) | 1.42(0.078) | 1.43(0.08) | 1.42(0.064) | - | – |
| **WHR (log$_{10}$)** | – | – | -0.01(0.04) | -0.06(0.033) | -0.07(0.042) | -0.06(0.045) | **Sex, Sus. × Sex** | – |
| **temperature (log$_{10}$˚C)** | – | – | 1.56(0.002) | 1.56(0.002) | 1.56(0.003) | 1.56(0.002) | - | – |
| **MAP (log$_{10}$ mmHg)** | – | – | 2.02(0.046)[§] | 1.98(0.051) | 1.97(0.063) | 1.98(0.046) | **Sex, Sus. × Sex** | – |
| **HR (log$_{10}$ bpm)** | – | – | 1.87(0.066) | 1.87(0.06) | 1.88(0.066) | 1.88(0.056) | - | – |
| **BR (log$_{10}$ cpm)** | – | – | 1.18(0.059) | 1.16(0.09) | 1.22(0.085) | 1.18(0.094) | - | – |
| **cholesterol (mg/dL)** | – | – | 177.46(32.58) | 175.91(29.9) | 186.24(30.0) | 183.89(39.8) | - | – |

Abbreviations used in this table are waist-to-hip ratio (WHR), mean arterial pressure (MAP), breathing rate (BR), heart rate (HR), and body mass index (BMI), standard deviation (SD), quartiles 1, 2 and 3 (Q1, Q2, Q3). Sex, Susceptibility (Sus.) and Sus. × Sex are the statistically significant factors and interaction of the 2-way ANOVA test. Significance levels of the paired Mann-Whitney U test are indicated as follows:

[¶] $p < 0.05$ compared to M+,

[#] $p < 0.05$ compared to F-, and

[§] $p < 0.05$ compared to F+.

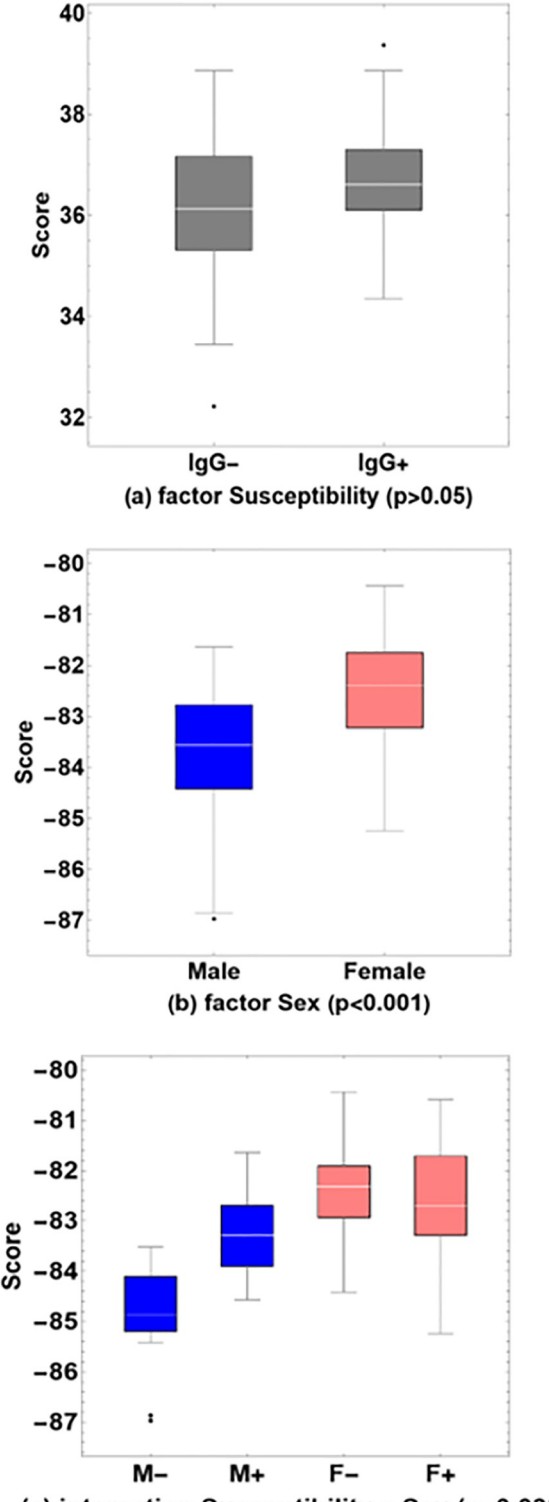

**Fig 2.** Box-whisker charts of MANOVA scores, for (a) the factor Susceptibility (p>0.05), (b) the factor Sex (p<0.001) and (c) the interaction Susceptibility × Sex (p = 0.033). Baseline values are shown for male and female participants who remained seronegative (M- and F-), or who were found to have been infected (M+ and F+), during a follow-up study six months later. Outliers are shown explicitly (black dots).

(1, 96) = 2.999, p = 0.067, partial η2 = 0.030; but there was a significant main effect for Sex, F (1, 96) = 5.848, p = 0.017, partial η2 = 0.57; and also a significant interaction between Susceptibility and Sex, F(1, 96) = 6.381, p = 0.013, partial $\eta^2$ = 0.062. There were no main effects for Susceptibility or Sex, nor interaction effects, on the remaining individual variables of glucose, cholesterol, BMI, BR, HR or temperature.

To further explore possible differences of individual continuous variables between pairs of groups, a Mann-Whitney U test was performed. We compared all possible pairs of the groups M-, M+, F- and F+ for each of the continuous variables, see Table 2 and Fig 3. The results indicated that values for uric acid (z = -2.121, p = 0.034), WHR (z = -3.403, p = 0.001) and MAP (z = -2.476, p = 0.013) are significantly larger for M- than for M+; values for urea (z = -2.830, p = 0.005), creatinine (z = -3.485, p<0.001), uric acid (z = -3.068, p = 0.002), WHR (z = -4.086, p<0.001) and MAP (z = -3.143, p = 0.002) are significantly larger for M- than for F-; values for urea (z = -3.544, p<0.001), creatinine (z = -2.617, p = 0.009), uric acid (z = -2.869, p = 0.004),

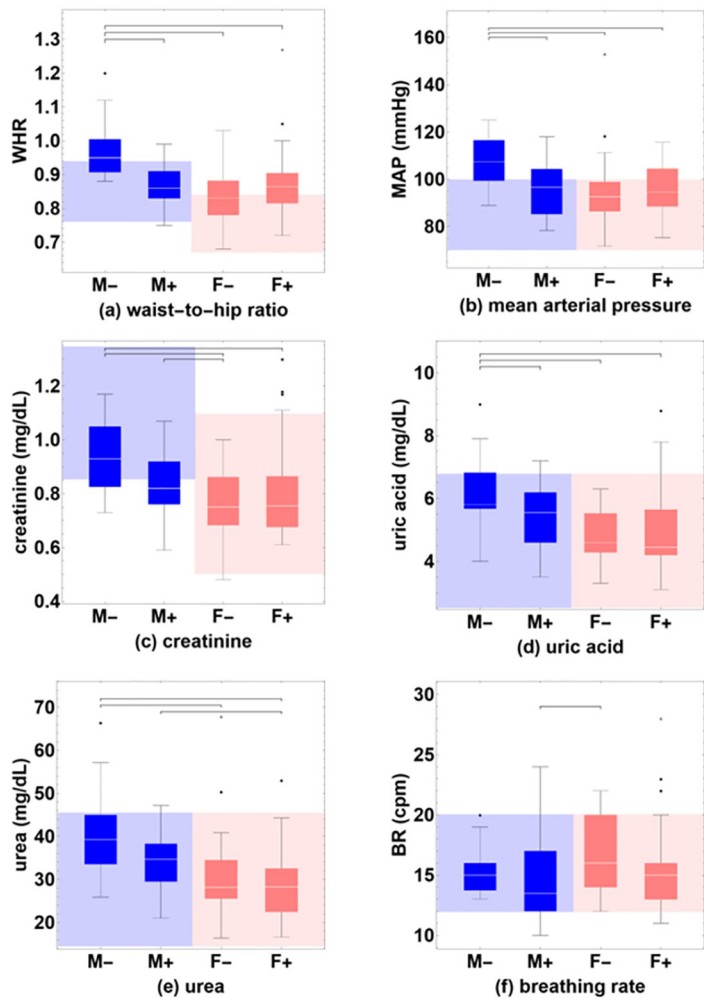

**Fig 3.** Box-whisker charts of values of the continuous variables that show significant pairwise differences between groups using the Mann-Whitney U test, (a) waist-to-hip ratio (WHR), (b) mean arterial pressure (MAP), (c) creatinine, (d) uric acid, (e) urea, and (f) breathing rate (BR). Baseline values are shown for male and female participants who remained seronegative (M- and F-), or who were found to have been infected (M+ and F+), during a follow-up study six months later. Normative ranges of reference values are indicated for male (blue-shaded background) and female participants (pink-shaded background). Outliers are shown explicitly (black dots).

WHR (z = -3.707, p<0.001) and MAP (z = -2.627, p = 0.009) are significantly larger for M-than for F+; values for creatinine (z = -2.532, p = 0.011) are significantly larger for M+ than for F- and values for BR (z = -2.289, p = 0.022) are significantly smaller for M+ than for F-; values for urea (z = -2.805, p = 0.005) are significantly larger for M+ than for F+. There are no statistically significant differences between F- and F+ for none of the individual dependent variables.

## Discussion

The original univariate analysis of the database of baseline parameters of the cohort did not show statistically significant differences between the IgG- and IgG+ groups for any of the continuous variables, and neither for the categorical variables except for hypertension [1]. In the present study, a multivariate analysis using 2-way MANOVA shows a borderline significance (p = 0.075) for the factor Susceptibility (IgG- vs. IgG+), suggesting that susceptibility is a systemic property that is better represented when considering the combined effect of all continuous variables together. The factor Sex (M vs. F) is highly significant (p<0.001), indicating that it is a dominant confounding factor that prevents the factor Susceptibility from reaching statistical significance independently. These findings confirm the widely acknowledged yet frequently overlooked reality of physiological differences between men and women concerning health-related parameters. This underscores the importance of separately analyzing groups of male and female participants [36, 37]. The inclusion of the interaction between both factors (M- vs. M+ vs. F- vs. F+) results in statistically significant group differences (p = 0.042) and indicates that there are systemic physiological states that are more susceptible to infection with SARS-CoV-2 than others, and that these states are modulated by sex.

A univariate 2-way ANOVA analysis shows that for the factor Susceptibility, only the variable urea is significant. For the factor Sex, half of the variables show significant differences: urea, creatinine, uric acid, waist-to-hip ratio, and mean arterial pressure. This again highlights that in the present dataset, it is inappropriate to analyze groups of male and female participants together, especially for physiological parameters where normative ranges of reference values are known to differ between the sexes. For the interaction between Susceptibility and Sex, three variables are significant: creatinine, mean arterial pressure, and waist-to-hip ratio.

To investigate which individual groups significantly deviate from other groups, we conducted a separate non-parametric pairwise univariate statistical test using the Mann-Whitney U test. There are no statistically significant differences between the groups of female participants (F- vs. F+) for any of the variables, and all values tend to be within the normative reference range. In contrast, there were statistically significant differences between the groups of male participants (M- vs. M+) for uric acid (p = 0.032), waist-to-hip ratio (p<0.001), and mean arterial pressure (p = 0.012). Uric acid is within the reference range for both groups, but the waist-to-hip ratio and mean arterial pressure variables are above the reference range for M- and within the range for M+.

One of the main findings of this study is the possible existence of different physiological states that are associated with varying degrees of susceptibility to infection with SARS-CoV-2. We recently published a series of contributions on physiological networks, where the central idea is that the human body is a complex system and that demographic, anthropomorphic, clinical, and laboratory variables do not exist in isolation but correlate into systemic physiological states that define the overall health condition [36, 38–42]. These physiological networks are influenced by sex [39, 42] and are disrupted by COVID-19 [39, 42].

The other main finding is that susceptibility to infection with SARS-CoV-2 is modulated by sex. While there were no differences in the physiological state and individual variables between the two groups of female participants (F- and F+), significant differences were observed

between the groups of male participants, which were either protected against (M-) or predisposed to (M+) infection. The individual variables that showed significant differences, uric acid, creatinine, waist-to-hip ratio, and mean arterial pressure, were consistently higher in M- compared to M+. All individual continuous variables were within the normal range for M+, but waist-to-hip ratio and mean arterial pressure were above the normal range for M-. These results are counter-intuitive because being overweight and having high blood pressure are known risk factors for the aggravated progression of COVID-19. Therefore, it appears there may be a specific set of risk factors for infection with SARS-CoV-2, and a different set of risk factors for an exacerbated progression of the COVID-19 disease. The present results show that the male sex exhibits specific risk factors for infection with SARS-CoV-2. Other studies have indicated that female COVID-19 patients have an increased risk to develop long COVID [43–45], whereas male patients have higher odds of requiring intensive treatment unit (ITU) admission or of having a fatal outcome [46]. Previously, we have explained the distinct evolution of COVID-19 for both sexes in terms of differences in the structure of their corresponding physiological networks [40, 42].

Our previous publication discussed susceptibility to infection with SARS-CoV-2 from the perspective of behavior. In particular, it was hypothesized that study participants who identified themselves as being vulnerable adapted their behavior to minimize exposure to the virus [1]. The present contribution raises the possibility that the behavior of the study participants may have been modulated by sex. An increasing body of literature emphasizes that distinct response patterns to the virus are evident between men and women, particularly manifesting in variations within self-care behaviors, and the greater compliance of women with preventive health behaviors has been contributed to personality traits of agreeableness and conscientiousness more typical of the female sex [47]. Such behavioral adaptations make up an important part of the non-pharmaceutical interventions to lower the viral inoculum and to reduce susceptibility to infection by SARS-CoV-2 [48]. Considering the higher adherence of women to preventive health behaviors, female participants within our cohort might have adjusted their behavior regardless of their perceived vulnerability, whereas for men, factors such as vulnerability and/or the presence of comorbidities might have played a more decisive role. An alternative interpretation could be differences in genetic susceptibility between the study groups, where variations in specific alleles can increase susceptibility or resistance, on the one hand to infection with SARS-CoV-2 [49, 50] and on the other hand to adverse health outcomes in COVID-19 disease [51–54]. In this context also, the results of the present contribution suggest that genetic susceptibility may be modulated by sex. However, the focus of the present contribution is physiological. Sex dimorphism affects susceptibility, severity, and progression of infections and disease [55]. Studies suggest that the inflammatory response may be different between men and women. It has been observed that in metabolic diseases, inflammatory mediators can be chronically or mildly activated [56], which could maintain a state of alertness of the innate immune system in a first line of defense against invasive pathogens. Whereas behavioral adaptations may have played a role in minimizing the exposure of the seronegative male participants of our cohort, it is also possible that the above-normal values for mean arterial pressure and waist-to-hip ratio that we observed in this group, and an associated state of low-grade inflammation, have resulted in a decreased susceptibility to infection.

Strengths and limitations. A possible limitation is the relatively small size of the study population. The calculation of the sample size indicated that the complete study population was sufficient for statistical analysis, but the size of the individual subgroups may be too restrictive to adequately reflect the more than seven thousand HCW employed by the HGMEL. A potential strength lies in the considerable homogeneity of our study population, with the vast majority vaccinated against COVID-19 using the same vaccine. Additionally, possible virus infections

were identified through two distinct methodologies: qPCR and IgG detection. We detected that an important confounding factor of our previous publication [1] was sex, which was resolved in the present contribution and which led to new results. We do not identify any other confounding factors as we are confident that possible infection and symptoms were correctly detected, without underreporting, and that all demographic, clinical and laboratory parameters were correctly registered.

## Conclusion

We examined the baseline data of healthcare workers to identify SARS-CoV-2 infection risk factors. We used multivariate statistics, such as MANOVA, to consider physiological factors. We found sex modulates susceptibility to infection. Men have different physiological states affecting their risk. In women, we did not find such distinctions. Men with normal arterial pressure and waist-to-hip ratio resulted to be more at risk. Being overweight and having hypertension worsen the development of COVID-19 disease, but may protect against initial SARS-CoV-2 infection.

## Supporting information

**S1 Data. Values for the categorical (sex, IgG) and continuous variables (glucose, urea, creatinine, uric acid, BMI, WHR, MAP BR, HR and temperature) of the cohort.**
(CSV)

## Author Contributions

**Conceptualization:** Octavio A. Lecona, América G. Arroyo-Valerio, Nallely Bueno-Hernández, José Damian Carrillo-Ruíz, Luis Ruelas, René Márquez-Franco, Eira Valeria Barrón, Galileo Escobedo, Elizabeth Ibarra-Coronado, Paola V. Olguín-Rodríguez, Antonio Barajas-Martínez, Ana Leonor Rivera, Ruben Fossion.

**Data curation:** Octavio A. Lecona, América G. Arroyo-Valerio, Nallely Bueno-Hernández, Luis Ruelas, René Márquez-Franco, Alejandro Aguado-García, Ana Leonor Rivera, Ruben Fossion.

**Formal analysis:** Octavio A. Lecona, Ana Leonor Rivera, Ruben Fossion.

**Funding acquisition:** Octavio A. Lecona.

**Investigation:** Octavio A. Lecona, América G. Arroyo-Valerio, Nallely Bueno-Hernández, José Damian Carrillo-Ruíz, René Márquez-Franco, Eira Valeria Barrón, Galileo Escobedo, Elizabeth Ibarra-Coronado, Ana Leonor Rivera, Ruben Fossion.

**Methodology:** Octavio A. Lecona, Alejandro Aguado-García, Elizabeth Ibarra-Coronado, Paola V. Olguín-Rodríguez, Antonio Barajas-Martínez, Ana Leonor Rivera, Ruben Fossion.

**Project administration:** América G. Arroyo-Valerio, José Damian Carrillo-Ruíz.

**Supervision:** América G. Arroyo-Valerio, Ana Leonor Rivera, Ruben Fossion.

**Validation:** América G. Arroyo-Valerio, Antonio Barajas-Martínez, Ruben Fossion.

**Visualization:** América G. Arroyo-Valerio, Ruben Fossion.

**Writing – original draft:** Octavio A. Lecona, América G. Arroyo-Valerio, Ruben Fossion.

**Writing – review & editing:** Octavio A. Lecona, América G. Arroyo-Valerio, Antonio Barajas-Martínez, Ruben Fossion.

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
