## [Decision Letter · Decision Letter 0]

31 Oct 2023

PONE-D-23-28208Risk factors contributing to infection with SARS-CoV-2 are modulated by sexPLOS ONE

Dear Dr. Fossion,

Thank you for submitting your manuscript to PLOS ONE. After careful consideration, we feel that it has merit but does not fully meet PLOS ONE’s publication criteria as it currently stands. Therefore, we invite you to submit a revised version of the manuscript that addresses the points raised during the review process.

We look forward to receiving your revised manuscript.

Kind regards,

Guadalupe Virginia Nevárez-Moorillón, Ph.D.

Academic Editor

PLOS ONE

Journal Requirements:

**Additional Editor Comments:**

Please consider the recommendations of both reviewers, in particular regarding the discussion and the limitations of the study.

Reviewers' comments:

Reviewer's Responses to Questions

**Comments to the Author**

1. Is the manuscript technically sound, and do the data support the conclusions?

Reviewer #1: Partly

Reviewer #2: Yes

2. Has the statistical analysis been performed appropriately and rigorously? 

Reviewer #1: Yes

Reviewer #2: Yes

3. Have the authors made all data underlying the findings in their manuscript fully available?

Reviewer #1: Yes

Reviewer #2: Yes

4. Is the manuscript presented in an intelligible fashion and written in standard English?

Reviewer #1: Yes

Reviewer #2: Yes

5. Review Comments to the Author

Reviewer #1: In a study involving healthcare workers during the initial phases of the COVID-19 outbreak in Mexico, researchers discovered that gender played a significant role in determining susceptibility to SARS-CoV-2 infection. Male participants exhibited varying physiological profiles that either heightened or diminished their vulnerability to the virus, whereas such distinctions were not evident among female participants. Of particular interest was the finding that uninfected males displayed values for mean arterial pressure and waist-to-hip ratio that fell outside the established norms. This observation suggests that risk factors commonly linked to severe COVID-19 outcomes may paradoxically confer a degree of protection against initial infection. This protection might be mediated through mechanisms related to chronic inflammation or the responsiveness of the innate immune system. While the study presents intriguing findings, further manuscript refinements are recommended:

Avoid including citations in the abstract.

Consider whether this study may be considered salami publication, as its results could potentially have been integrated with those of Bueno-Hernández et al. (2022).

Describe the confounding factors identified in this study and outline the measures taken to control them.

Provide information about the vaccination status of the individuals at the 6-month follow-up and assess the potential protective effects.

Discuss the potential implications of this study for the development of long COVID, making reference to these relevant studies (DOI: 10.3390/jcm11020413, DOI: 10.1080/03007995.2022.2081454, DOI: 10.3390/jcm11247314) that highlights a possible association between sex and long COVID.

Reviewer #2: Thanks for the opportunity to review the manuscript by Octavio A. Lecona and his collaborators.

The authors explored the database of the cohort's demographic, physiological, and laboratory parameters recorded at baseline to identify potential risk factors for infection with SARS-CoV-2 six months later at a follow-up evaluation. They hypothesized that a multivariate statistical analysis, such as MANOVA, maybe a more appropriate approach.

The manuscript is well-written, and the study design is well done. In general, the authors made a reasonable interpretation of their results. The statistical tool selected is described and explained. I really have only minor comments to clarify for the authors.

In the introduction section, it is stated: "...whereas several participants of the IgG- group were hypertensive." Please indicate the hypertensive grade/severity in those participants suffering from it and its importance in your results.

Also, in the introduction, the paragraph beginning with "The results of our previous publication were based on a preliminary dataset of n=110 ..." is somewhat wordy; I suggest reformulating it as follows:

The results of our previous publication were based on a preliminary dataset of n=110 participants that was constructed with the parameters and the evaluation results that were available at that time, including the continuous variables of breathing rate (BR), body temperature, weight, body mass index (BMI), waist circumference, and hip circumference Also, blood glucose, urea, creatinine, uric acid and cholesterol, and the categorical variables of obesity, type 2 diabetes, and hypertension. The present contribution uses an updated dataset that includes additional parameters such as heart rate (HR), mean arterial pressure (MAP), and waist-to-hip ratio (WHR) but only considers n=100 participants with complete data. It has been argued before that susceptibility to infection may be systemic and physiological, apart from a local or cellular property (Caputo et al., 2022).

The discussion is well-written and includes main topics related to leading findings. However, the authors obviate that much research has been done on the Mexican population regarding genetic susceptibility in this topic; a quick search in PubMed can help them in this regard, for example, PMIDs: 35892712, 35395388, 37108839, 35967349, and many others; please consider discussing the lack of assessing the potential genetic factors in your research. A brief paragraph could be essential to add.

The conclusion section is excessive; the authors should make a concise conclusion according to their results, avoiding being repetitive with the discussion and methods sections.

Minor comment:

Please homogenize the term SARS-CoV-2. In some places, it is wrong, for example, in the first paragraph of the introduction.

6. PLOS authors have the option to publish the peer review history of their article (what does this mean?). If published, this will include your full peer review and any attached files.

Reviewer #1: No

Reviewer #2: No

---

## [Author Response · Author response to Decision Letter 0]

5 Jan 2024

See attached DOCX file with detailed response to the editor and the referees

---

## [Editor Report · Decision Letter 1]

16 Jan 2024

Risk factors contributing to infection with SARS-CoV-2 are modulated by sex

PONE-D-23-28208R1

Dear Dr. Fossion,

We’re pleased to inform you that your manuscript has been judged scientifically suitable for publication and will be formally accepted for publication once it meets all outstanding technical requirements.

Kind regards,

Guadalupe Virginia Nevárez-Moorillón, Ph.D.

Academic Editor

PLOS ONE
---

## [Editor Report · Acceptance letter]

1 Feb 2024

PONE-D-23-28208R1 

PLOS ONE

Dear Dr. Fossion, 

I'm pleased to inform you that your manuscript has been deemed suitable for publication in PLOS ONE. Congratulations! Your manuscript is now being handed over to our production team.

Kind regards, 

on behalf of

Dr. Guadalupe Virginia Nevárez-Moorillón 

Academic Editor

PLOS ONE